# Use of Online Food Delivery Services to Order Food Prepared Away-From-Home and Associated Sociodemographic Characteristics: A Cross-Sectional, Multi-Country Analysis

**DOI:** 10.3390/ijerph17145190

**Published:** 2020-07-17

**Authors:** Matthew Keeble, Jean Adams, Gary Sacks, Lana Vanderlee, Christine M. White, David Hammond, Thomas Burgoine

**Affiliations:** 1UKCRC Centre for Diet and Activity Research (CEDAR), MRC Epidemiology Unit, University of Cambridge School of Clinical Medicine, Box 285 Institute of Metabolic Science, Cambridge Biomedical Campus, Cambridge CB22 0QQ, UK; jma79@medschl.cam.ac.uk (J.A.); tb464@medschl.cam.ac.uk (T.B.); 2Global Obesity Centre, Deakin University, Geelong VIC 3220, Australia; gary.sacks@deakin.edu.au; 3School of Nutrition, Université Laval, Quebec, QC G1V 0A6, Canada; lana.vanderlee@fsaa.ulaval.ca; 4School of Public Health and Health Systems, Faculty of Applied Health Sciences, University of Waterloo, Waterloo, ON N2L 3G1, Canada; c5white@uwaterloo.ca (C.M.W.); david.hammond@uwaterloo.ca (D.H.)

**Keywords:** fast food, food accessibility, food delivery, food environment, online food delivery services, out-of-home food, public health, takeaway food

## Abstract

Online food delivery services like Just Eat and Grubhub facilitate online ordering and home delivery of food prepared away-from-home. It is poorly understood how these services are used and by whom. This study investigated the prevalence of online food delivery service use and sociodemographic characteristics of customers, in and across Australia, Canada, Mexico, the UK, and the USA. We analyzed online survey data (*n* = 19,378) from the International Food Policy Study, conducted in 2018. We identified respondents who reported any online food delivery service use in the past 7 days and calculated the frequency of use and number of meals ordered. We investigated whether odds of any online food delivery service use in the past 7 days differed by sociodemographic characteristics using adjusted logistic regression. Overall, 15% of respondents (*n* = 2929) reported online food delivery service use, with the greatest prevalence amongst respondents in Mexico (*n* = 839 (26%)). Online food delivery services had most frequently been used once and the median number of meals purchased through this mode of order was two. Odds of any online food delivery service use were lower per additional year of age (OR: 0.95; 95% CI: 0.94, 0.95) and greater for respondents who were male (OR: 1.50; 95% CI: 1.35, 1.66), that identified with an ethnic minority (OR: 1.57; 95% CI: 1.38, 1.78), were highly educated (OR: 1.66; 95% CI: 1.46, 1.90), or living with children (OR: 2.71; 95% CI: 2.44, 3.01). Further research is required to explore how online food delivery services may influence diet and health.

## 1. Introduction

According to global estimates from 2016, 11% of men and 15% of women were living with obesity, which has been associated with multiple co-morbidities [1,2]. Whilst the drivers of obesity are complex, the role of excess calorie intake through consumption of food prepared away-from-home has been recognized in previous research [3,4,5]. Food prepared away-from-home is often energy dense, high in fat and salt, and less healthy than food prepared at home, and more frequent consumption has been associated with elevated bodyweight [6,7,8,9,10].

Food prepared away-from-home is typically served ready to consume and has become a major contributor to overall dietary intake [11,12]. In the USA, for example, food prepared away-from-home accounted for over 50% of total food expenditure in 2018 [13]. Traditionally, this food may have been purchased through ‘conventional’ modes of order whereby customers would visit food outlets in-person or contact food outlets directly to place orders before collection or delivery. Third-party platforms that facilitate online ordering and delivery, referred to throughout as ‘online food delivery services’, provide an alternative mode of order that appears to have grown in popularity [14]. Whilst business models vary, online food delivery services typically operate as intermediaries between customers and food outlets [15]. Customers place orders through online platforms, their orders are forwarded to food outlets where meals are cooked, and once ready, meals are delivered to customers by couriers working for the food outlet or the online food delivery service [14,16].

In 2020, prominent online food delivery services Just Eat (including subsidiaries) and Uber Eats, were available in 13 countries, Deliveroo was available in 12 countries, and Grubhub was established in many cities across the USA [17,18,19,20]. Online food delivery service availability has been forecast to increase, which could lead to greater use. In turn, this could increase the purchase and consumption of food prepared away-from-home [21]. To our knowledge, there is currently a limited understanding about the nutritional quality of food items sold through online food delivery services. Nonetheless, given that food sold through online food delivery services is primarily prepared in existing food outlet facilities [15], it may have a similar nutrient profile to food prepared away-from-home ordered in conventional ways. As such, online food delivery services could contribute to excess calorie intake and adverse health outcomes [6,7,22]. Accordingly, interventions to reduce online food delivery service use or to improve the nutritional quality of food that is available, may be called for in the future.

Previous research into online food delivery services is limited. A narrative review identified business reports stating that convenience and choice of food outlet were potential drivers of online food delivery service use, supporting findings from Malaysia and Indonesia [16,23]. A further study investigated the availability of food outlets through an online food delivery service in one city in each of Australia, the Netherlands, and the USA [24]. In each city, a diverse range of food types were available and the number of food outlets that were available differed by area level deprivation. To date, the prevalence and frequency of online food delivery service use and the sociodemographic characteristics of online food delivery service customers have not been investigated, and thus remain poorly understood. Understanding how often online food delivery services are used and the sociodemographic characteristics of current online food delivery service customers will establish a baseline against which future use can be compared, allow any future interventions to be targeted towards frequent users, serve as an indicator of potential public health harm and of the need for further research.

In this study, we aimed to describe the prevalence and frequency of online food delivery service use, investigate associations between online food delivery service use and sociodemographic characteristics, and describe how online food delivery service customers used other modes of order to purchase food prepared away-from-home, in and across five upper-middle or high-income countries.

## 2. Materials and Methods 

### 2.1. Data Collection

We used cross-sectional data from the International Food Policy Study (IFPS), conducted in Australia, Canada, Mexico, the UK, and the USA in November 2018. Data collection methods have been described elsewhere [24]. Briefly, data were collected via self-completed online surveys from adults aged 18 years or over, recruited through Nielsen Consumer Insights Global Panel and their partners’ panels. Panelists were screened for eligibility and quota requirements based on device screen size, age, and sex. Email invitations containing links to an online survey in national languages were sent to a random sample of eligible panelists in each country. Respondents provided consent prior to survey completion. The IFPS was reviewed by and received ethics clearance through a University of Waterloo Research Ethics Committee (ORE# 21460).

### 2.2. Measures

All respondents were asked: “During the past 7 days, how many meals did you get that were prepared away-from-home in places such as restaurants, fast food or takeaway places, food stands, or from vending machines?”. A similar question has been asked in previous research [25,26]. Respondents who had purchased at least one meal prepared away-from-home reported the number of meals ordered: “using a food delivery service (e.g. country specific examples) and delivered”, “directly from a restaurant and delivered”, “at a restaurant/food outlet within 5 minutes of your home”, and “at a restaurant/food outlet more than 5 minutes away from your home”. Country-specific examples of online food delivery services available in each country included Uber Eats (all countries), Just Eat (Canada, Mexico, UK), Deliveroo (Australia, UK), Foodora (Australia), SkipTheDishes (Canada), and Grubhub (USA). In our analyses, we collapsed “at a restaurant/food outlet within 5 minutes of your home” and “at a restaurant/food outlet more than 5 minutes away from your home” into a single category: ‘directly from food outlets in-person’.

We included sex, age, ethnicity, education, body mass index (BMI), and living with children aged under 18 years as independent variables. Age was reported in years (continuous). Ethnicity was reported as the group that best described racial or ethnic backgrounds. We dichotomized responses into ‘majority’ (white, predominantly English speaking or not indigenous) and ‘minority’ (all other responses). Education was reported as the highest level completed. We categorized respondents as having: ‘low’ (high school completion or lower), ‘medium’ (some post-high school qualifications), or ‘high’ (university degree or higher) levels of education, and used this variable as a marker of socioeconomic status [27]. Height and weight were reported in either metric or imperial units. We calculated BMI (kg/m^2^) and grouped respondents by World Health Organization categories: ‘underweight’ (BMI < 18.5), ‘normal weight’ (BMI 18.5–24.9), ‘overweight’ (BMI 25.0–29.9), or ‘obesity’ (BMI ≥ 30) [28]. We collapsed the ‘underweight’ and ‘normal weight’ categories into a ‘not overweight’ category (BMI < 25.0), and as individuals with greater BMI may not always report their height and weight, we included respondents with missing data for this variable and categorized them as ‘missing’ [29,30]. Living with children aged under 18 years was reported as a binary variable.

### 2.3. Study Sample

In total, 22,824 respondents completed the online survey. We excluded respondents with missing data for variables of interest (except for BMI), when the total number of meals purchased away-from-home and the number of meals purchased through each mode of order summed did not match, or when the total number of meals purchased away-from-home in the past 7 days exceeded 21 (*n* = 2164). We considered 21 to be the maximum number of meals that could be purchased away-from-home based on daily consumption of three meals in the past 7 days. The final analytical sample included 19,378 respondents.

### 2.4. Statistical Analysis

To reduce non-response and selection bias, we applied post-stratification sample weights. Weights were constructed using population estimates from the census in each country based on age, sex, region, ethnicity (except in Canada) and education (except in Mexico) [24,31].

In each country, we determined the prevalence of online food delivery service use by identifying respondents who reported that they had used an online food delivery service at least once in the past 7 days. For these ‘online food delivery service customers’ we identified the frequency of online food delivery service use and calculated the number and proportion of all meals purchased away-from-home for each mode of order (‘online food delivery services’, ‘directly from food outlets for delivery’ and ‘directly from food outlets in-person’). For respondents who had purchased at least one meal prepared away-from-home directly from food outlets for delivery or in-person but had not used an online food delivery service (‘non-online food delivery service customers’), we calculated the number and proportion of all meals purchased away-from-home ‘directly from food outlets for delivery’ and ‘directly from food outlets in-person’.

In analyses, we used online food delivery service use as our dependent variable. As data were not normally distributed, we dichotomized respondents into any online food delivery service use in the past 7 days or not. We used Pearson’s χ^2^ to compare differences in sociodemographic characteristics of online food delivery service customers in each country. To investigate associations between our dependent variable and sex, age, ethnicity, education, BMI, and living with children aged under 18 years, across all countries combined, we used logistic regression following a sequential modelling strategy. Model 0 was unadjusted, Model 1 was adjusted for all independent variables except education, to investigate variation by individual-level socioeconomic status, and Model 2 was maximally adjusted [32,33]. To investigate differences in online food delivery service use between countries, we used separate, maximally adjusted, logistic regression models with each country as the reference category. We investigated differences in prevalence of online food delivery service use and independent variables between countries by adding a two-way interaction term (country x independent variable) to separate maximally adjusted logistic regression models and used post-estimation Wald tests to determine interaction term significance. When interaction terms were significant, we stratified analyses by country. We used Stata version 15.1 (StataCorp LLC., College Station, TX, USA) to complete analyses in 2019, with a significance threshold of *p* < 0.05 used throughout.

## 3. Results

Amongst our sample, 78% (*n* = 15,093) had purchased at least one meal prepared away-from-home in the past 7 days; 15% (*n* = 2929) had used an *online* food delivery service at least once, and 63% (*n* = 12,163) had purchased food prepared away-from-home directly from food outlets for delivery or in-person, but had not used an online food delivery service.

### 3.1. Sociodemographic Characteristics

Overall, more than half of our sample were female or identified with an ethnic majority, most had low education, over 20% were living with obesity, the median age was 47 years, and less than 30% lived with children aged under 18 years (Appendix A). Overall, more than half of the 2929 online food delivery service customers were male, identified with an ethnic majority, were highly educated, or were living with children aged under 18 years, while around 40% were living with overweight or obesity, and the median age was 33 years (Table 1). Sociodemographic characteristics of respondents that had purchased at least one meal prepared away-from-home directly from food outlets for delivery or in-person, but had not used an online food delivery service (*n* = 12,163), are shown in Appendix A.

### 3.2. Meals Purchased Away-From-Home

Around half of respondents that reported any online food delivery service use in the past 7 days, had used this mode of order once (Appendix A). Table 2 reports the modes of order used to purchase meals prepared away-from-home. Overall, online food delivery service customers ordered a median of two meals prepared away-from-home through an online food delivery service, which represented 36% of all meals purchased away-from-home. Online food delivery service customers also ordered a median of one meal directly from food outlets for delivery and two meals directly from food outlets in-person. Overall, the median number of meals that non-online food delivery service customers ordered directly from food outlets for delivery was two, which was the same as the median number of meals ordered directly from food outlets in-person.

### 3.3. Sociodemographic Correlates of Online Food Delivery Service Use

Sociodemographic correlates of any online food delivery service use in the past 7 days from unadjusted and partially adjusted models are reported in Appendix A. Figure 1 reports findings from the maximally adjusted model. Overall, there were greater odds of online food delivery service use amongst respondents who were male (OR: 1.50; 95% CI: 1.35, 1.66), that identified with an ethnic minority (OR: 1.57; 95% CI: 1.38, 1.78), those who lived with children aged under 18 years (OR: 2.71; 95% CI: 2.44, 3.01), or had high (versus low) levels of education (OR: 1.66; 95% CI: 1.46, 1.90). Odds of online food delivery service use were lower per additional year of age (OR: 0.95; 95% CI: 0.94, 0.95) and there were no differences by BMI category.

### 3.4. Between-Country Variation

The greatest prevalence of any online food delivery service use in the past 7 days was amongst respondents in Mexico (*n* = 895 (26%)). Respondents in Canada had lower odds of online food delivery service use compared to respondents in all other countries, whilst respondents in the UK and Mexico had greater odds compared to respondents in all other countries (Table 3). Amongst online food delivery service customers in Australia, Mexico, and the USA, the median number of meals ordered through online food delivery services per person, was two, whereas in Canada and the UK, the median number, per person, was one (Table 2).

There were significant between-country interactions. The association between online food delivery service use in the past 7 days and each of age (*p* < 0.0001), living with children aged under 18 years (*p* = 0.037), sex (*p* < 0.0001), and education (*p* < 0.0001) varied between countries (Appendix A). Figure 2, Figure 3, Figure 4 and Figure 5 report country-stratified findings. Odds of online food delivery service use in the past 7 days were lower per additional year of age amongst respondents in all countries. Respondents who lived with children aged under 18 years had greater odds of online food delivery service use in all countries, with the strongest association observed amongst respondents in the USA (OR: 3.22; 95% CI: 2.49, 4.20). There was no difference in odds of online food delivery service use by sex amongst respondents in Mexico (OR: 1.02; 95% CI: 0.85, 1.23), whereas males in all other countries had greater odds of online food delivery service use. Respondents with high (versus low) levels of education had greater odds of online food delivery service use in all countries except the UK (OR: 0.87; 95% CI: 0.67, 1.13).

## 4. Discussion

### 4.1. Summary of Findings

To our knowledge, this is the first study in the published international literature that has examined the prevalence and frequency of online food delivery service use and identified sociodemographic characteristics of online food delivery service customers. Our findings from multiple countries provide knowledge about the individual and wider contextual factors that may relate to online food delivery service use. Overall, 15% of respondents across Australia, Canada, Mexico, the UK, and the USA reported online food delivery service use in the past 7 days, however, almost two thirds of respondents had purchased food prepared away-from-home directly from food outlets but had *not* used an online food delivery service. Online food delivery services were most frequently used once in the past 7 days. Overall, online food delivery service customers ordered a median of two meals through an online food delivery service, and the median proportion of all meals purchased away-from-home ordered through an online food delivery service was more than 30%. Respondents who were male, younger, with higher education, lived with children aged under 18 years, or that identified with an ethnic minority had greater odds of online food delivery service use. Respondents in Mexico and the UK had greater odds of online food delivery service use compared to respondents in other countries, and whilst correlates of online food delivery service use were similar in each country, the strength of associations varied.

### 4.2. Interpretation of Findings and Further Research

As the first study to investigate the prevalence and frequency of online food delivery service use in and across multiple countries, we are unable to conclude that the levels we identified are relatively high or low. Nonetheless, our findings regarding the modes of order used to purchase food prepared away-from-home provide novel insight into how multiple ways of purchasing food prepared away-from-home may coexist. When having food delivered, those who reported any online food delivery service use in the past 7 days appeared to favor this mode of order compared to ordering directly from food outlets. Our observation could support the suggestion that online food delivery services have the capacity to disrupt conventional and established purchasing formats within food retail, which in turn, could influence how individuals interact with the built food environment [34]. However, a high proportion of online food delivery service customers reported that they also visited food outlets in-person, indicating that the traditional practice of visiting neighborhood food outlets persisted regardless of online food delivery service use. Therefore, promotion of healthier neighborhood food environments, for example through the use of urban planning or ‘zoning’ continues to be a potentially effective public health intervention [35]. Importantly, using multiple modes of order to purchase food prepared away-from-home may lead to greater total consumption, increased risk of excess weight and adverse health outcomes [36,37]. The full extent to which using multiple modes of order, and in-particular online food delivery service use, increases consumption of food prepared away-from-home, is unclear without longitudinal data.

Consistent with our finding that men had greater odds of online food delivery service use, men reportedly purchase food prepared away-from-home more frequently and cook at home less than women [38,39]. It is unclear how reasons for purchasing food prepared away-from-home might differ based on mode of order used, and how these reasons may vary by sex.

Respondents that identified with an ethnic minority had greater odds of online food delivery service use. Analyses of data from the National Health and Nutrition Examination Survey completed in the USA indicated that black respondents cooked at home less frequently than other groups [40]. However, further research from the USA [41] and UK [42] concluded that individuals that identified with an ethnic minority allocated more time to home food preparation and consumed more home cooked food than individuals that identified with an ethnic majority. Online food delivery service use could reduce home cooking, which might have implications for the overall diet quality of customers. Whilst it is possible to meet dietary guidelines through consumption of food prepared away-from-home, it may be more difficult and more expensive than through food prepared at home [42,43], and bound by the types of food outlet available [44].

In our study, online food delivery service customers were likely to be younger, have higher education, or live with children aged under 18 years. Similarly, marketing companies and online food delivery services suggest that individuals with these sociodemographic characteristics often report online food delivery service use [45]. Older individuals may be disinclined to order food online due to lacking familiarity with technology and a loyalty towards conventional modes of order, whilst individuals who are younger, highly educated, or parents, often report having limited time and may purchase food prepared away-from-home to offset pressure stemming from having limited time resources [46,47,48,49]. As previously described, reasons for using one mode of order over another are currently unclear [50]. Future research should engage with online food delivery service customers to better understand their reasons for online food delivery service use.

Analysis of the UK’s National Diet and Nutrition Survey identified that living with obesity was associated with greater consumption of food from fast-food outlets but not restaurants or cafés [51]. In our study, online food delivery service use was not associated with weight status. To some extent, this may be due to our cross-sectional study design and the simultaneous measurement of our exposure (online food delivery use) and outcome (weight status). However, it is also possible that this reflects the potential for online food delivery services to offer food from different types of food outlets, including restaurants, which may offer healthier food than is traditionally served away-from-home [45]. In our analysis it was not possible to disaggregate online delivery service use by the type of food outlet that meals were ordered from. Future research investigating which food outlets are ordered from when using online food delivery services, and the nutritional composition of foods sold, would provide greater insight into associations between food delivery service use and weight status. This understanding would serve to inform the need for development of public health interventions.

The prevalence of online food delivery service use, the proportion of all meals prepared away-from-home purchased through online food delivery services, and the number of meals purchased directly from food outlets in-person by non-online food delivery service customers, were each greatest for respondents in Mexico. Together, these findings may reflect cultural norms aligned with frequent purchase of food prepared away-from-home in this country [52].

Individuals with greater access to food outlets through online food delivery services could be inclined to use them more frequently. This may explain plans from Just Eat, branded as SkipTheDishes, to increase the number of food outlets in Canada who are signed up to accept orders through their platform [19]. Indeed, our finding that respondents from Canada had lower odds of online food delivery service use compared to respondents in all other countries could indicate that there is currently limited access to food outlets through this mode of order. Future research could investigate the extent to which access to food outlets signed up to accept orders through online food delivery services is associated with online food delivery service use.

Sociodemographic characteristics of online food delivery service customers were similar between countries, however, the strength of associations varied. Notably, higher education was associated with greater odds of online food delivery service use in all countries except the UK. Food outlets signed up to accept orders through online food delivery services in the UK may not sell food that accommodates the needs of individuals with higher education, possibly limiting use. The type of food available through online food delivery services in the UK is currently unclear. Whilst the UK may be different, amongst food outlets signed up to accept orders through an online food delivery service in Australia, the Netherlands, and the USA, common food labels used to describe the type of food sold included ‘Burgers’, ‘Pizza’, and ‘Italian’, with ‘Healthy’ food labels less common [53]. However, labels selected by food outlets may not always reflect the food they sell and the nutritional quality of food available through online food delivery services remains unclear. Given the apparent lack of ‘Healthy’ food choices, further work to develop an understanding about how well self-selected labels reflect the types of food that outlets sell, and the nutritional quality of this food, is warranted.

### 4.3. Limitations

This study represents the most comprehensive description of online food delivery service use to date. Nonetheless, the findings are subject to limitations, including those common to survey-based research. Respondents were recruited using nonprobability-based sampling. Thus, findings are not necessarily nationally representative. We applied post-stratification sample weights to improve representativeness, yet respondents in Mexico had higher levels of education than census estimates and average BMI scores were lower than national averages for respondents in all countries [24]. Recruitment may have been biased towards individuals with internet access. In 2016, however, internet penetration rates ranged between 67% (Mexico) and 93% (Australia), with rates of 88% or higher in Canada, the UK, and the USA [54].

Analyses were based on cross-sectional data, limiting the ability to draw causal inference. Additionally, data were self-reported and collected through online surveys. Social desirability bias may have led to the number of meals purchased away-from-home, online food delivery service use, and body weight being under-reported. This risk may have been reduced by use of online surveys that offer respondents a sense of anonymity when reporting sensitive information [55,56]. Finally, we used education as our marker of socioeconomic status which may not be internationally comparable [39,57].

It is possible that the global COVID-19 pandemic has accelerated changes in consumer behavior with regards to use of online modes of order [58]. At least in terms of the research contexts studied here, individuals that may have previously visited food outlets in-person to purchase food prepared away-from-home are likely to have found that this option has been restricted, and may therefore have adopted online modes of order. Whilst there is much uncertainty, it is possible that short-term changes in consumer behavior persist long term. Research is required to fully understand short- and long-term changes in online food delivery service and in-person food outlet use.

## 5. Conclusions

We found that 15% of adults across Australia, Canada, Mexico, the UK, and the USA had purchased food prepared away-from-home through online food delivery services in the past 7 days. Online food delivery service use was associated with being male, from an ethnic minority, younger, highly educated, or living with children aged under 18 years. Sociodemographic characteristics of online food delivery service customers were consistent across countries, yet there was variation in the strength of associations. Norms surrounding the purchase of food prepared away-from-home, stressors on time that limit the opportunity for home meal preparation, and the number and type of food outlets that can be accessed through online food delivery services may vary internationally and could help explain observed differences between countries. Whilst we identified sociodemographic characteristics of online food delivery service customers, which is important information for future intervention development, further research is needed to understand the extent to which use of an online food delivery service contributes to overall purchasing and consumption of food prepared away-from-home, whether online food delivery services are used in place of, or in addition to, traditional modes of order, and associated implications for public health.

## Figures and Tables

**Figure 1 ijerph-17-05190-f001:**
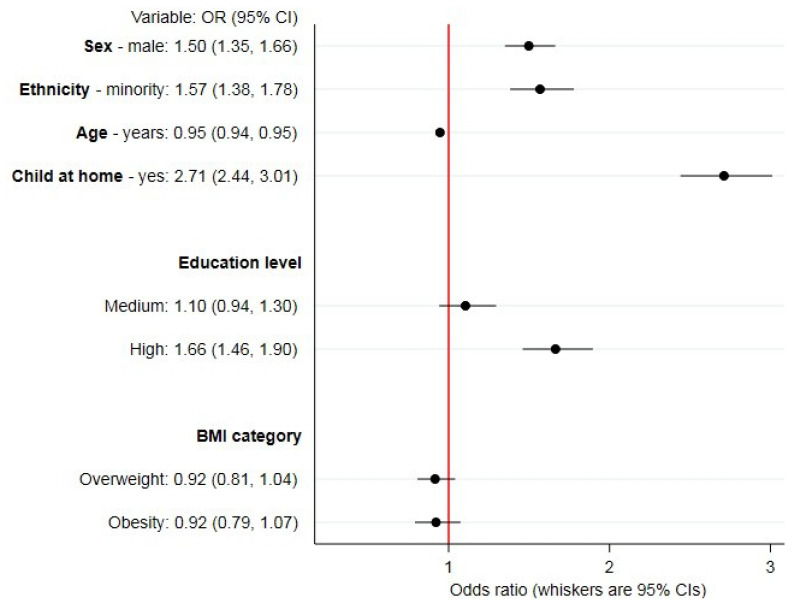
Associations between prevalence of any online food delivery service use in the past 7 days and sociodemographic characteristics (*n* = 19,378). Data are from 2018, collected through the International Food Policy Study, analyzed using adjusted logistic regression. Note: reference groups: ethnicity—majority, education level—low, body mass index (BMI) category—not overweight.

**Figure 2 ijerph-17-05190-f002:**
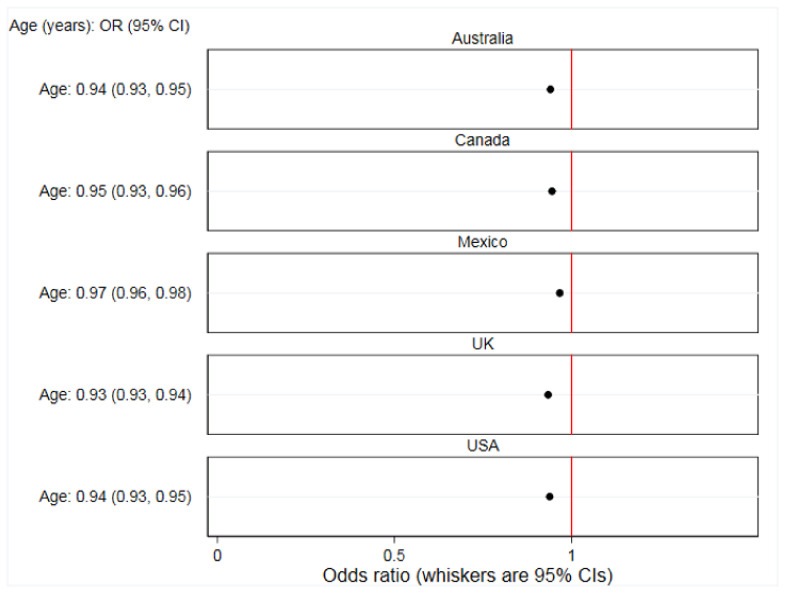
Associations between prevalence of any online food delivery service use in the past 7 days and age (*n* = 19,378). Data are from 2018, collected through the International Food Policy Study, analyzed using country-stratified adjusted logistic regression.

**Figure 3 ijerph-17-05190-f003:**
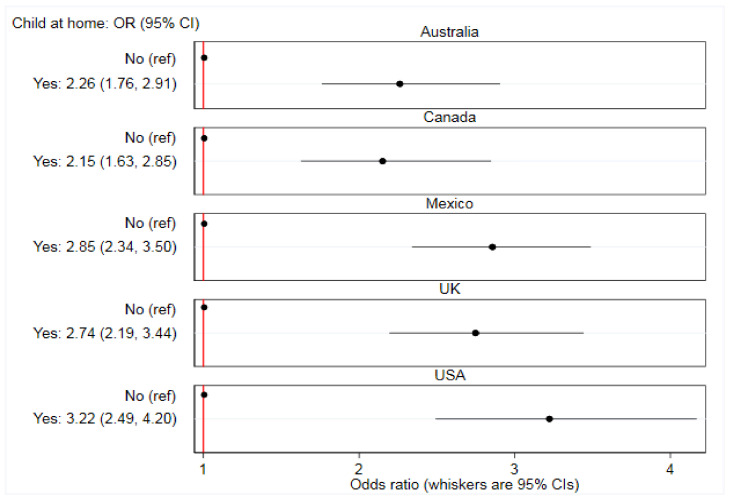
Associations between prevalence of any online food delivery service use in the past 7 days and living with a child aged under 18 years (*n* = 19,378). Data are from 2018, collected through the International Food Policy Study, analyzed using country-stratified adjusted logistic regression.

**Figure 4 ijerph-17-05190-f004:**
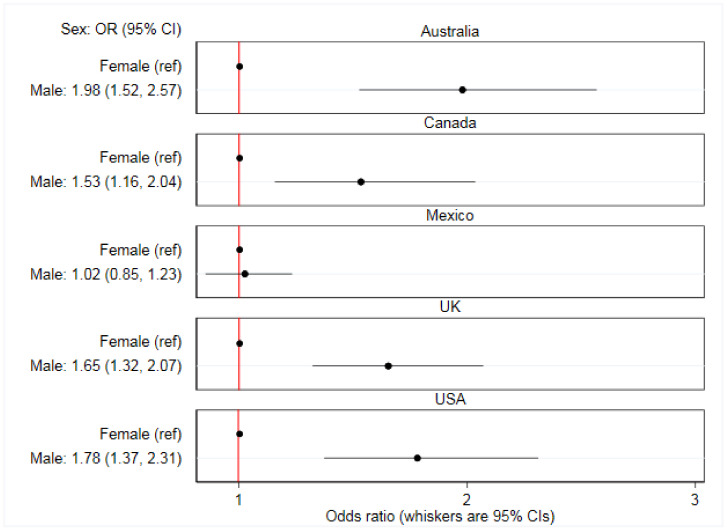
Associations between prevalence of any online food delivery service use in the past 7 days and sex (*n* = 19,378). Data are from 2018, collected through the International Food Policy Study, analyzed using country-stratified adjusted logistic regression.

**Figure 5 ijerph-17-05190-f005:**
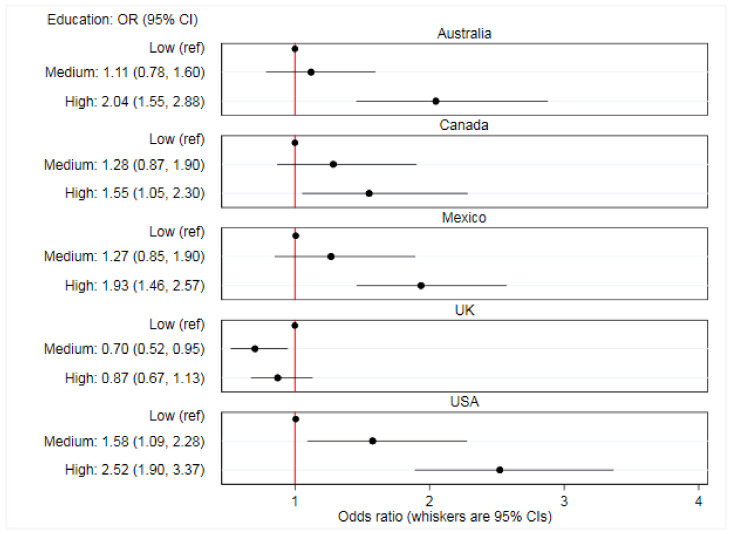
Associations between prevalence of any online food delivery service use in the past 7 days and education level (*n* = 19,378). Data are from 2018, collected through the International Food Policy Study, analyzed using country-stratified adjusted logistic regression.

**Table 1 ijerph-17-05190-t001:** Sociodemographic characteristics of online food delivery service customers (*n* = 2929) ^a^.

	Australia (*n* = 3578)		Canada (*n* = 3698)		Mexico (*n* = 3515)		UK (*n* = 4694)		USA (*n* = 3893)		Total (*n* = 19,378)		*p* Value for Difference ^b^
Online food delivery service customers ^c^	498	(13.9)	327	(8.8)	895	(25.5)	747	(15.9)	461	(11.9)	2929	(15.1)	*p >* 0.0001
Variable													
Sex													*p >* 0.0001
Male	305	(61.1)	196	(60.0)	433	(48.3)	422	(56.4)	273	(59.2)	1629	(55.6)	
Ethnicity													*p >* 0.0001
Majority	310	(32.1)	201	(61.3)	662	(73.9)	570	(76.3)	259	(56.1)	2001	(68.3)	
Age (years)													
Median (IQR)	31	(25–40)	33	(26*–*41)	34	(27*–*42)	32	(25*–*41)	33	(26*–*38)	33	(26*–*41)	
Education													*p >* 0.0001
Low	136	(27.4)	95	(28.9)	119	(13.3)	320	(42.8)	177	(38.3)	846	(28.9)	
Medium	133	(26.6)	110	(33.7)	90	(10.1)	171	(22.9)	35	(7.6)	538	(18.4)	
High	229	(46.0)	122	(37.4)	686	(76.6)	257	(34.4)	250	(54.4)	1545	(52.7)	
BMI (kg/m^2^)													*p >* 0.0001
Not overweight (≤24.9)	255	(51.2)	164	(50.2)	420	(46.9)	321	(42.9)	206	(44.6)	1366	(46.6)	
Overweight (25.0–29.9)	118	(23.7)	77	(23.4)	265	(29.6)	150	(20.1)	135	(29.2)	744	(25.4)	
Obesity (≥30.0)	52	(10.3)	60	(18.2)	145	(16.1)	106	(14.2)	77	(16.7)	439	(15.0)	
Missing	73	(14.7)	27	(8.2)	66	(7.4)	170	(22.8)	44	(9.5)	380	(13.0)	
Child < 18 years in home													*p >* 0.0001
Yes	226	(45.4)	131	(40.0)	639	(71.4)	364	(48.7)	240	(51.9)	1600	(54.6)	

Note: ^a^—Unless specified, data reported as n (%). ^b^—p values from Pearson’s χ2 test. ^c^—Online food delivery service customers had purchased at least one meal prepared away-from-home through an online food delivery service in the past 7 days.

**Table 2 ijerph-17-05190-t002:** Modes of order used in the past 7 days to purchase meals prepared away-from-home.

	Australia (*n* = 3578)		Canada (*n* = 3698)		Mexico (*n* = 3515)		UK (*n* = 4694)		USA (*n* = 3893)		Total (*n* = 19378)		*p* Value for Difference ^a^
Online food delivery service customer’s ^b^: n (%)	498	(13.9)	327	(8.8)	895	(25.5)	747	(15.9)	461	(11.9)	2929	(15.1)	
Online food delivery services ^c^													*p >* 0.0001
Number of meals	2.0	(1.0–2.0)	1.0	(1.0–2.0)	2.0	(1.0–3.0)	1.0	(1.0–2.0)	2.0	(1.0–3.0)	2.0	(1.0–3.0)	
Proportion (%)	40.0	(50.0–57.1)	40.0	(25.0–66.7)	33.3	(23.1–50.0)	50.0	(33.3–100.0)	33.3	(25.0–50.0)	35.7	(25.0–50.0)	
Directly from food outlets for delivery ^c^													*p >* 0.0001
Number of meals	0.0	(0.0–1.0)	0.0	(0.0–1.0)	1.0	(0.0–2.0)	0.0	(0.0–1.0)	1.0	(0.0–2.0)	1.0	(0.0–2.0)	
Proportion (%)	0.0	(0.0–28.6)	0.0	(0.0–28.6)	25.0	(0.0–40.0)	0.0	(0.0–25.0)	20.0	(0.0–33.3)	16.7	(0.0–33.3)	
Directly from food outlets in–person ^c^													*p >* 0.0001
Number of meals	2.0	(0.0–3.0)	1.0	(0.0–3.0)	2.0	(1.0–4.0)	1.0	(0.0–2.0)	2.0	(1.0–4.0)	2.0	(0.0–3.0)	
Proportion (%)	40.0	(0.0–60.0)	40.0	(0.0–54.5)	40.0	(20.0–52.4)	33.3	(0.0–50.0)	40.0	(20.0–57.1)	40.0	(0.0–50.0)	
Non–online food delivery service customers ^d^: n (%)	2188	(61.2)	2420	(65.4)	2396	(68.2)	2439	(52.0)	2721	(69.9)	12163	(62.8)	
Directly from food outlets for delivery ^c^													*p >* 0.0001
Number of meals	1.0	(1.0–2.0)	1.0	(1.0–2.0)	2.0	(1.0–4.0)	1.0	(1.0–2.0)	2.0	(1.0–3.0)	2.0	(1.0–3.0)	
Proportion (%)	66.7	(50.0–100.0)	100.0	(40.0–100.0)	75.0	(50.0–100.0)	100.0	(50.0–100.0)	100.0	(40.0–100.0)	83.3	(50.0–100.0)	
Directly from food outlets in-person ^c^													*p >* 0.0001
Number of meals	2.0	(1.0–3.0)	2.0	(1.0–3.0)	3.0	(2.0–4.0)	2.0	(1.0–3.0)	2.0	(1.0–4.0)	2.0	(1.0–3.0)	
Proportion (%)	100.0	(100.0–100.0)	100.0	(100.0–100.0)	100.0	(66.7–100.0)	100.0	(100.0–100.0)	100.0	(100.0–100.0)	100.0	(100.0–100.0)	

Note: ^a^—p value from Pearson’s χ^2^ test. ^b^—Online food delivery service customers had purchased at least one meal prepared away-from-home through an online food delivery service in the past 7 days. ^c^—Data reported as median (Interquartile Range (IQR)) number of meals, and median (IQR) proportion of all meals purchased away-from-home, per person. ^d^—Non-online food delivery service customers had purchased at least one meal prepared away-from-home directly from food outlets but not through an online food delivery service, in the past 7 days.

**Table 3 ijerph-17-05190-t003:** Associations between country and prevalence of online food delivery service use in the past 7 days ^a^.

Country	OR ^b^	95% CI ^c^	
Australia (reference)	-	-	
Canada	0.65	0.54	0.78
Mexico	1.21	1.03	1.43
UK	1.39	1.18	1.64
USA	0.85	0.72	1.02
Australia	1.55	1.29	1.87
Canada (reference)	-	-	
Mexico	1.88	1.58	2.25
UK	2.15	1.79	2.57
USA	1.32	1.10	1.59
Australia	0.82	0.69	0.97
Canada	0.53	0.45	0.63
Mexico (reference)	-	-	
UK	1.14	0.98	1.33
USA	0.70	0.60	0.82
Australia	0.72	0.61	0.85
Canada	0.47	0.39	0.56
Mexico	0.88	0.75	1.02
UK (reference)	-	-	
USA	0.61	0.52	0.73
Australia	1.17	0.98	1.40
Canada	0.76	0.63	0.91
Mexico	1.43	1.22	1.67
UK	1.63	1.37	1.93
USA (reference)	-	-	

Note: ^a^—Each country used as a reference in separate adjusted logistic regression models. ^b^—Odds ratio. ^c^—95% confidence intervals.

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
