# Peer review of "Use of Online Food Delivery Services to Order Food Prepared Away-From-Home and Associated Sociodemographic Characteristics: A Cross-Sectional, Multi-Country Analysis"

_ijerph, 2020, doi:10.3390/ijerph17145190_

Round 1

Reviewer 1 Report

The present paper describe the prevalence and frequency of online food delivery service use, investigate associations between online food delivery service use and sociodemographic characteristics, and describe how online food delivery service customers used other modes of order to purchase food prepared away-from-home, in and across five upper-middle or high-income countries.

The manuscript is interesting and deals with a very important and uptodate topic.

Due to the very recent outbreak for COVID-19 that deeply influenced lifestyle and nutritional habits, I think that a comment on this would be useful. In this regard it may be useful to quote this recent manuscript ("Quarantine during COVID-19 outbreak: Changes in diet and physical activity increase the risk of cardiovascular disease". Doi: 10.1016/j.numecd.2020.05.020.)

Do this pandemic changed the food delivery services?

Line 285 In our study, online food delivery service use was not associated with weight status.

Authors need to explain the limitation of the study methods related to this specific point.

Reviewer 2 Report

Is there an extra line in Table 3 at the end? USA reference is shown twice.

USA (reference) - -

Please use a high resolution image for figure 2. They are helpful but very difficult to read.

Please add a few sentences about online ordering and food quality in Australia. If there is zero research, please specify that. The discussion does a good job on touching the other three countries in explaining the results with relevance.

Very interesting research! Well done!

Reviewer 3 Report

In this manuscript “Use of online food delivery services to order food prepares away-from-home and associated sociodemographic characteristics: a cross-sectional, multi-country analysis” by Matthew Keeble and co-authors presented their work on the odds of any online food delivery service use in the past 7-days differed by sociodemographic characteristics through adjusted logistic regression.

There are a few points that have to be addressed and modify the manuscript before publishing:

  1. Keywords are not in alphabetical order.
  2. The knowledge gap of the present work is not clearly mentioned.
  3. Tables: Authors are suggested to include footnote describing the statistical representation.
  4. Authors are suggested to mention some of the examples of effective knowledge translation strategies in the conclusions section.
  5. Authors are suggested to strengthen the importance and relevance of the present results in the discussion section.
  6. Significance of the study should be highlighted.
  7. References should be cited by following journal-style/format.
  8. Need to check for typographical errors, plagiarism, punctuation, and grammar throughout the manuscript.

Round 2

Reviewer 3 Report

In this manuscript “Use of online food delivery services to order food prepares away-from-home and associated sociodemographic characteristics: a cross-sectional, multi-country analysis” by Matthew Keeble and co-authors presented their work on the odds of any online food delivery service use in the past 7-days differed by sociodemographic characteristics through adjusted logistic regression.

I think this paper is now improved but the following points should be checked once again.

  1. I think it is a known fact the food cooked away from home and online orders have adverse effects on the health of any socio-economic population. I wish to know in which way the manuscript help the public to lead a healthy lifestyle?!
  2. References should be checked once again if that is cited by following journal-style/format.
  3. Need to once again check for typographical errors, plagiarism, punctuation, and grammar throughout the manuscript.
